# *Amphipterygium adstringens* (Schltdl.) Schiede ex Standl (Anacardiaceae): An Endemic Plant with Relevant Pharmacological Properties

**DOI:** 10.3390/plants11131766

**Published:** 2022-07-02

**Authors:** Mireya Sotelo-Barrera, Marcela Cília-García, Mario Luna-Cavazos, José Luis Díaz-Núñez, Angélica Romero-Manzanares, Ramón Marcos Soto-Hernández, Israel Castillo-Juárez

**Affiliations:** Posgrado en Botánica, Colegio de Postgraduados, Texcoco 56230, Mexico; sotelo.mireya@colpos.mx (M.S.-B.); cilia.marcela@colpos.mx (M.C.-G.); mluna@colpos.mx (M.L.-C.); diaz.jose@colpos.mx (J.L.D.-N.); dahly@colpos.mx (A.R.-M.); msoto@colpos.mx (R.M.S.-H.)

**Keywords:** *Amphipterygium adstringens*, cuachalalate, Anacardiaceae, long-chain phenols, anacardic acids, triterpenes, anticancer activity

## Abstract

Diseases, such as cancer, peptic ulcers, and diabetes, as well as those caused by drug-resistant infectious agents are examples of some of the world’s major public health problems. *Amphipterygium adstringens* (Schltdl.) Schiede ex Standl is an endemic tree to Mexico. Its stem bark has been used medicinally since pre-Hispanic times, but in recent decades it has been scientifically proven that it has properties that help counteract some diseases; extracts with organic solvents of the plant are outstanding for their anticancer, gastroprotective, and antimicrobial properties; terpenes and long-chain phenols have been identified as the main active compounds. Currently, overharvesting is causing a sharp reduction in natural populations due to an increase in demand for the stem bark by people seeking to improve their health and by national and transnational companies seeking to market it. Because of the growing interest of the world population and the scientific community, we reviewed recent studies on the bioactive properties of *A. adstringens*. Through the orderly and critical compendium of the current knowledge of *A. adstringens*, we provide a reference for future studies aimed at the rational use and protection of this valuable endemic natural resource.

## 1. Introduction

*Amphipterygium adstringens* (Schltdl.) Schiede ex Standl is an endemic tree to Mexico, popularly known as “cuachalalate”, and whose stem bark has been traditionally used to treat different diseases [1]. It is one of Mexico’s most intensively harvested medicinal plants due to its high local commercial demand [2].

The term “cuachalalate” (in Spanish) comes from the Nahuatl roots “cuahuitl”, which means tree, while “chachacuachtic” means rough so the meaning would be a rough tree [3]. Moreover, the Nahuatl word “cuauchachalatli” is made up of “cuáhuitl”, which means tree, and “chachalatli”, which alludes to the “chachalaca” (bird of the genus *Ortalis* sp. with a noisy song), so its meaning would be “chachalacas tree” [4].

Traditionally, the stem bark has been used to treat more than 25 ailments [1], including emerging and global public health diseases, such as cancer, peptic ulcers, and infectious diseases.

Pharmacological research on *A. adstringens* began in the mid-20th century, but it has been in recent decades that the interest of the scientific community has increased. Currently, it has been scientifically proven that the stem bark has anticancer [5,6], anti-inflammatory [7], gastroprotective [8], antimicrobial properties [9], and wound healing activity [10]. Moreover, terpenes and phenolic compounds have been identified as some of the main active metabolites [7,8,9,10].

However, because of its health benefits, uncontrolled debarking, excessive trade, and loss of habitat, in addition to the alteration of climatic patterns, the plant’s survival has been put at risk and its natural distribution has been considerably reduced [11,12]. Currently, *A. adstringens* is classified in the IUCN Red List of Threatened Species^TM^ as *Vulnerable* [12]. For this reason, the design of strategies to achieve sustainable use and long-term conservation of the species is urgently required [13].

This review, motivated by the scientific community’s interest in this plant resource, aimed to make a compendium of the current knowledge of *A. adstringens*, its biology, its pharmacological properties, and of the available sustainability studies.

The methodology used was based on searching electronic databases (PubMed, Web of Science, Scopus, ScienceDirect, Google Scholar, Research Gate, Mexican Institute of Industrial Property (IMPI), National Institute of Statistics and Geography (INEGI), World Intellectual Property Organization (WIPO), technical manuals, and historical documents). The keywords used were: *Amphipterygium adstringens*, *Juliana adstringens*, cuachalalate, anacardic acids, masticadienonic acid, and 3 α-hydroxymasticadienonic acid. The last access was in May 2022.

## 2. Taxonomic Classification and Morphology

In 1843, Schlechtendal described a specimen from the State of Morelos and named it *Hypopterygium adstringens* Schlecht, but later changed it to the genus *Juliania* since *Hypopterygium* belongs to a genus of mosses. In 1906, Hemsley proposed the formation of the Julianiaceae family with two genera, *Orthopterygium* and *Juliania*, which are dioecious trees or shrubs with resinous branches and deciduous leaves grouped at the end of the branch [14].

In 1923, Standley changed the name of the genus *Juliania* to *Amphipterygium* [15]. However, later studies allowed it to be placed in the Anacardiaceae Family since it was observed that the shape of the style and the stigmas of the female flowers were reminiscent of this family [15]. Moreover, the presence of resinous channels, the structure of the ovary and the ovule [16], the anatomy of its wood [17], and the chemical composition were sufficient elements to place it in Anacardiaceae [18]. The current classification is shown in Table 1.

Physical characteristics: The height of the trees is 6 to 10 m, with an average diameter of 40 cm at the middle of the stem (Figure 1a and Figure 2a). The stem is curved, and the branches are thick, ascending with sympodial branching. The outer bark begins with the periderm, a thin, smooth, dark-gray layer with numerous lenticels (Figure 1b). The inner bark is pinkish and exudes creamy white resin with a pungent odor [19]. (Figure 1c–e and Figure 2b,c).

The leaves are arranged in a spiral, clustered at the end of the branches, odd-pinnate, and measure 6 to 13 cm. The petiole is composed of three to five opposite leaflets without a petiole, and the terminal leaflet is the longest. They are ovate or elliptic with a crenate margin, apex, acute base, opaque-yellowish-green on the upper side, and grayish-green on the lower side (Figure 2 and Figure 3). 

The male flowers form clustered panicles in the axils of the leaves, up to 15 cm long, tomentose with a pedicel up to 3 mm long, or actinomorphic sessile with a diameter of 3 to 4 mm. The perianth consists of five to seven segments, 1.5 to 2 mm long-linear, acutely tomentose. It has five to seven stamens, 1 to 1.5 mm long, with a short filament, an oblong and tomentose anther, and an absent ovary (Figure 3). 

The female flowers are found solitary in the axils of the new leaves, on flattened and elongated peduncles 1 cm long and 3 to 4 mm wide. The receptacles are globular, 3 mm long, with five acute teeth, containing an ovary with two semi-united carpels, seminiferous, unilocular, and pubescent, with three recurved stigmatic branches with pubescent style and stigma (Figure 2d and Figure 3). 

The fruit is a 3 to 4 cm dry indehiscent samara, with persistent stigmas on flattened pedicels that continue to grow to form a kind of wing, yellowish to reddish, fused with one or two 5 mm seeds (Figure 2e and Figure 3). The fruit is distributed by the wind beginning in October (dry season), and some fruits remain on the tree until the next rainy season [20].

## 3. Distribution

Tropical Dry Forest (TDF) occupies 38% of the surface in Mexico [21]. This ecosystem is characterized by drought of up to eight months and annual rainfall of less than 1500 mm [22].

*Amphipterygium adstringens* is an endemic species of Mexico (although there are some records of collection in Guatemala) distributed in the TDF [19], mainly in the vegetation type known as low deciduous forest [23] (Figure 4). 

The distribution of *A. adstringens* covers the states of Nayarit, Jalisco, Colima, Michoacán, Guerrero, Oaxaca, Mexico, Puebla, and Morelos [19,24]. 

This species grows in warm sub-humid climate with 400 mm rainfall and a mean annual temperature of 24 °C on slopes greater than 30% and parent material of volcanic origin [19]. In the dry season (November–April) the trees lose their leaves, while in the rainy season (May–October) they show abundant foliage [25] (Figure 1 and Figure 2).

## 4. Traditional Uses and Ethnopharmacology

The stem bark of the tree is currently used to treat more than 25 ailments, including gastritis, peptic ulcers, skin diseases, and wounds [11,26,27,28]. In the XVI century Francisco Hernández in his work *History of the Plants of New Spain*, mentions it as “chalalactli” and the use of the ground stem bark to “reduce tumors” [29]. Moreover, in the *Essay on Mexican Materia Medica* of 1832, it is mentioned as “cuauchalalá”, and cooking the bark in water to “firm the teeth”, as well as spraying the infusion to heal animal skin sores [30] (Figure 5a). In the *Mexican Pharmacopoeia formed and published by the Pharmaceutical Academy of the Capital of the Republic* of 1846, it is mentioned as “cuanchalalate” or “cuanchalalá”, whose stem bark is applied topically to heal wounds [31].

In 1944, Maximino Martínez makes a brief botanical description and refers to the “cuachalalate” stem bark as an astringent, with anticancer and antimalarial properties [32]. It is also attributed with curative properties against cancer and typhoid fever [27], as well as an antidiabetic, blood cholesterol reducer, and kidney stone and gallstone destroyer, and as a cure for “urine disease” (urinary tract infections or cystitis) and whooping cough (pertussis) and to “purify the blood” [33]. Similarly, it reduces ovarian inflammation and regulates blood glucose levels [34].

Because of its antiseptic properties, it is used to treat wounds, pimples, animal bites, sores, bumps, insect bites [32], hair loss, wounds, burns, ulcers, mouth sores, “skin spots” (melasma), varicose ulcers, and gangrene [35].

The main modes of administration are ingestion of the aqueous extract and topical application of powdered stem bark (Figure 5b,c). For skin diseases, the powdered bark is applied directly to the skin (Figure 5b,c). In some cases, the bark resin treats pimples and abscesses [28]. The stem bark is macerated or boiled in water until it turns color and is consumed as daily water to relieve gastric ulcers [36].

## 5. Phytochemical Composition

The main classes of compounds present in the stem bark of *A. adstringens* include terpenes and sterols, among which **3-epioleanolic acid**, **β-sitosterol**, **masticadienonic acid**, and **3α-hydroxymasticadienonic acid** stand out [37,38,39] (Figure 6).

The *Mexican Herbal Pharmacopoeia* mentions the use of 3α-hydroxymasticadienonic acid as a specific marker for *A. adstringens* [31]. However, masticadienonic acid and oleanolic acid are also suggested as quality control markers for herbal products [40].

Other compounds that have been identified in the stem bark are instipolinacic acid [41], cuachalalic acid [37], oleanonic acid, schinol (masticadienolic acid), 3β-hydroxymasticadienolic acid, acotillone, terpene 11 (3α-hydroxy-11α,12α-epoxy-oleanane-28,13β-olide), and terpene 12 (3β-hydroxy-11α,12α-epoxy-oleanane-28,13β-olide) [42]. It should be noted that the above compounds had already been reported in other plant species, but Makino et al. described five new tirucallane-type terpenes produced in the stem bark of *A. adstringens*: terpene 1 (3α-hydroxy-6-oxo-7,24*Z*-tirucalladien -26-oic acid), terpene 2 (3,7-dioxo-8, 24*Z*-tirucalladien-26-oic acid), terpene 3 (3α-hydroxy-7-oxo-8,24*Z*-tirucalladien-26-oic acid), terpene 4 (7,11-dioxo-3α-hydroxy-8,24*Z*-tirucalladien-26-oic acid), and terpene 5 (3,8-dioxo-7β-hydroxy-7,9-cycro-7,8-seco-24*Z*tirucalladien-26-oic acid) (Figure 6) [42].

Furthermore, although the most popular form of administration is aqueous extract, its chemical composition has been barely studied. The presence of saponins in the aqueous extract is suggested by the formation of foam when shaken vigorously (Figure 5c). However, only one steroidal saponin has been isolated, which, by comparison with a reference sample, was identified as **sarsasapogenin**, which is glycosylated with D-glucose and D-rhamnose residues [43]. 

Long-chain phenols have also been identified in the stem bark; of these, anacardic acids stand out. Although this class of molecules has been placed in other members of the Anacardiaceae Family, the anacardic acid mixture of *A. adstringens* is mainly constituted of saturated chains of 15 to 19 carbon atoms. Some of those that have been reported are 6-pentadecyl salicylic acid (C15:0), 6-hexadecyl salicylic acid (16:0), 6-heptadecyl salicylic acid (17:0), 6-nonadecenyl salicylic acid (19:0), 6-[15’(Z)-nonadecenyl]-salicylic acid (19:1∆15) [9,10,33,44,45], and 6[16’(Z)-nonadecenyl]-salicylic acid (C19:1∆16) [46].

Similarly, an anacardic aldehydes mixture consisting of 6-octadecyl salicylaldehyde (18:0), 6-eicosyl salicylaldehyde (20:0), and 6-docosyl salicylaldehyde (22:1) has been identified [44]. The presence of naphthalene, 3-dodecyl-1,8-dihydroxy-2-naphthoic acid, was also reported [47].

Moreover, the HPLC (high-performance liquid chromatography) and GC/MS (gas chromatography/mass spectrometry) analysis of the ethanolic extract identified the presence of other compounds, such as catechins, cardanols, β-terpineol, catechol, naringenin, pinocembrin, and fatty acids [48]. Likewise, non-hydrolysable tannins with a catechol nucleus were identified [43].

## 6. Pharmacological Properties

Studies have been carried out mainly on the extracts with organic solvents of the stem bark, in which terpenes and anacardic acids have been identified as the main active compounds (Figure 7). The different pharmacological trials reported in the literature to date are described below:

### 6.1. Anticancer

The first reports related to the anticancer properties of *A. adstringens* appeared in the mid-twentieth century as part of screening programs to identify plants with antitumor properties [49]. A later study reported that the methanolic extract of the stem bark reduced tumor growth 54% in a mammary adenocarcinoma murine model. The antineoplastic activity was identified mainly in polar fractions (aqueous) and was related to glycosylated steroidal saponins [43]. It should be noted that an adverse effect of some fractions of low polarities (petroleum ether and chloroform) that stimulated tumor growth 57 to 113% [43] was also recorded. Finally, sarsapogenin (Figure 7) was identified (its glycosylated form contains D-glucose and D-rhamnose residues), but no studies were conducted to corroborate its antitumor activity individually [43].

Furthermore, although there are few studies on the anticancer properties of the aqueous extract of the stem bark, its ability to restore the immunological parameters of immunocompromised mice with L5178Y lymphoma has been reported. Moreover, oral administration of the extract for ten days (10 and 100 mg/kg) induced an increase in the immune cell response and the proliferation of splenocytes in the animals [50].

Most of the anticancer studies of the stem bark have focused on the analysis of extracts with organic solvents and their components. In this regard, the cytotoxic activity of the methanolic extract (4–27.5 μg/mL) against various human cancer cells, such as melanoma, colon, and ovarian adenocarcinoma, was identified [5]. 

Moreover, cytotoxic activity of the ethanolic extract was reported to decrease the viability of L-1210 leukemia cells, and seven terpenes (schinol, masticadienonic acid, oleanolic acid, acotillone, and terpenes 2, 11, and 12) with moderate cytotoxic activity (IC_50_ from 20 to 40 μg/mL) were identified [42] (Figure 7).

Masticadienonic and 3α-hydroxymasticadienonic acids were isolated from the hexane extract, which inhibited the growth of five human cancer cell lines and stimulated nitric oxide production in macrophages [7]. Recently, it was reported that 3α-hydroxymasticadienonic acid induced the maturation of dendritic cells, which present antigens and are involved in the T lymphocyte activation [9,50]. Interestingly, the coadministration of dendritic cells with triterpene to melanoma mice decreased tumor growth and increased the area of cell death in tumors [9].

Other identified compounds with anticancer properties are the so-called anacardic acids. The first report described 6-nonadecenyl salicylic acid (C19:0), which was isolated from the hexane extract and whose oral administration of 10 mg/kg in mice showed cytotoxic effects on peripheral blood cells [45].

Subsequent studies have focused on analyzing 6-pentadecyl salicylic acid (C15:0), one of the primary anacardic acids in the mixtures that is commercially available (Calbiochem). This compound showed cytotoxic activity on human gastric cancer and leukemia cell lines. Furthermore, it exhibited genotoxic activity by increasing the frequency of micronucleate cells, DNA degradation, and caspase-8-dependent apoptosis. However, it showed low toxicity on normal human peripheral blood mononuclear cells and mouse bone marrow polychromatic erythrocytes [51].

Immunosuppression is a secondary effect of various antineoplastic agents; therefore, identifying anticancer molecules that can stimulate the immune system is crucial [52,53]. In this regard, C15:0 exhibited immunomodulatory properties by promoting the adaptive immune system and increasing macrophage activation through the phosphorylation of mitogen-activated protein kinase and nuclear factor-kB. Moreover, peritoneal macrophages increased the secretion of interleukin-6, tumor necrosis factor-α (TNF-α), and nitric oxide. Similarly, it stimulated their migration and phagocytic activity, while interleukin-4 and interleukin-10 decreased. In addition, it did not show toxicity, nor did it alter the proportion of T lymphocytes (auxiliary and cytotoxic), natural killer cells (NK), and macrophages [54]. In contrast, the intravenous administration of C15:0 in mice (three doses of 2 mg/Kg every seven days) increased the production of macrophages and NK and proinflammatory cytokines, such as TNF-α and interleukins 2, 12, 6, and 1β [55].

Furthermore, unlike other chemotherapeutic agents, such as taxol, cisplatin, and 5-fluorouracil, C15:0 does not induce myelosuppression or leukopenia [52,53]. In this regard, the intravenous administration of C15:0 (2 mg/Kg/48 h/21 days) reduced breast cancer tumor cells in immune-competent mice [52]. The induction of apoptosis regulated by caspase-8 was recorded; it also contributed to antitumor immunity by increasing blood immune cells in lymph nodes and bone marrow without affecting lymphocytes or immune cell subpopulations [52]. Finally, C15:0, combined with chemotherapeutic agents, also decreased myelosuppression and leukopenia caused by Taxol [52], cisplatin, and 5-fluorouracil [53].

### 6.2. Antiulcer and Gastroprotective

One of the leading traditional uses of the stem bark is due to its gastroprotective and antiulcer properties [56,57]. In this regard, it was reported that oral administration (3 times every 24 h for 5 days) of the aqueous extract (4 and 8% weight/volume) reduced death and perforation of the duodenum induced by indomethacin and histamine in Wistar rats [58]. It also prevented the formation of gastric ulcers, although reduction in gastric juice secretion was not recorded [58].

A bio-guided assay of the methanolic extract of the stem bark previously defatted with hexane was performed in a subsequent study. Oral administration of the methanolic extract and various fractions rich in triterpenes reduced the formation of gastric ulcers induced by absolute ethanol in rats [59]. In this study, 3α-hydroxymasticadienonic acid was identified as one of the active compounds, but its activity was lower than that of the active fractions and bismuth subsalicylate, which was used as a positive control [59].

Similarly, the methanolic extract showed a gastroprotective effect (72.5% at 300 mg/kg) similar to that of omeprazole (50–89.7% at 130 mg/kg) in a model of gastric injury induced by diclofenac [8]. Moreover, a dichloromethane fraction was obtained from the methanolic extract, where compounds with gastroprotective activity in ethanol-induced gastric lesions were identified [60]. The most active was 3-epioleanolic acid with action similar to that of the positive control carbenoxolone (88.8%), followed by 3α-hydroxymasticadienonic acid (69.8%) and β-sitosterol (42.5%). It was found that nitric oxide participates in the gastroprotective mechanism, which increases the synthesis of endogenous prostaglandins and sulfhydryl [60].

Recently, the synergistic gastroprotective effect of 3α-hydroxymasticadienonic acid in combination with diligustilide (dimeric phthalide isolated from *Ligusticum porteri* J.M. Coult & Rose), administered orally in a model of gastric injury induced by indomethacin, was reported. It is suggested that the gastroprotective mechanism of action involves increased prostaglandin synthesis and reduced leukocyte infiltrate, TNF-α, and leukotriene B4 [61].

### 6.3. Anti-Inflammatory

Differences in the aqueous and hexane extract activity have been reported in two models of acute inflammation induced by chemical agents. Topically applied hexane extract (1 mg/ear) reduced by 50% ear edema caused by 12-*O*-tetradecanoylphorbol-13-acetate, while the aqueous extract, administered orally (31 mg/kg), decreased the foot edema of animals induced by carrageenan by 82% [7]. However, the participation of masticadienonic and 3α-hydroxymasticadienonic acids was identified in both models. It has been suggested that, as part of the mechanism of action, it inhibits pro-inflammatory mediators since terpenes strongly reduce nitric oxide production [7].

Furthermore, the anti-inflammatory effect of the ethanolic extract has been demonstrated in a model of colitis induced by dextran sulfate sodium in mice. Oral administration of the extract at 200 mg/kg for ten days reduced colonic inflammation and increased animal survival by up to 80%. Moreover, it reduced weight loss and increased the activity of antioxidant enzymes, such as superoxide dismutase and glutathione peroxidase. Likewise, it presented immunomodulatory properties since it decreased the levels of the cytokines IFN-γ, IL-1β, and TNF-α [48].

### 6.4. Antimicrobial

*Mycobacterium tuberculosis* is an intracellular bacterium that is difficult to eradicate and whose problem is aggravated by the presence of multi-drug resistant strains [66]. In this regard, it was reported that the CH_2_Cl_2_/MeOH (1:1) extract at 0.5 mg/mL inhibited 95% of *M. tuberculosis* growth. Masticadienonic acid and 3α-hydroxymasticadienonic acid were identified as active compounds, with MIC of 0.064 and 0.032 mg/mL, respectively [47].

The methanolic extract inhibited the growth of *Streptococcus mutans* (MIC 0.069 mg/mL) and *Porphyromonas gingivalis* (MIC 0.082 mg/mL); four anacardic acids were identified as active compounds (C19:1, C15:1, C19:0, and C15:0) and masticadienonic acid, with an MIC of 0.007 to 0.150 mg/mL for *S. mutans* and 0.012 to 0.175 mg/mL for *P. gingivalis* [46]. Similarly, the methanolic extract exhibited bactericidal activity against other oral pathogens, such as *Aggregatibacter actinomycetemcomitans*, *Candida albicans*, and *Candida dubliniensis,* with MIC from 0.4 to 63 mg/mL and MBC from 1.6 to 63 mg/mL [5].

The *Helicobacter pylori* bacterium is the primary etiological agent of active chronic gastritis, peptic ulcer, and gastric cancer [67]. It was reported that the aqueous extract (MIC 5 mg/mL) and the methanolic extract (MIC 0.25 mg/mL) of the stem bark inhibit its growth [63]. Similarly, petroleum ether extract and anacardic acid mixture (C15:0 [46.8%], C16:0 [7.2%], C17:0 [29.9%], C19:0 [7.5%], and C19:1 [8.6%]) as the main active compounds, with a MIC of 0.01 mg/mL inhibited its growth. In the mechanism of action, a lytic effect on the bacterial membrane was identified [9].

Moreover, the hexane extract and anacardic acid mixture at subinhibitory concentrations inhibited quorum sensing in *Chromobacterium violaceum* and *Pseudomonas aeruginosa* [64,65]. In *C. violaceum*, 0.055 mg/mL of extract inhibited 91.6% of violacein pigment production, while the anacardic acid mixture did so by 94% with 0.166 mg/mL [64]. 

In *P. aeruginosa*, the anacardic acid mixture reduced the production of virulence factors (anti-virulence), such as pyocyanin (86% at 0.2 mg/mL), rhamnolipids (91% at 0.5 mg/mL), and elastolytic activity (75% at 0.5 mg/mL) [64]. At the same time, the methanolic extract showed moderate inhibitory activity against *Trypanosoma cruzi*, which is the etiologic agent of Chagas disease [62]. 

### 6.5. Wound Healing

In a murine excision model, it was shown that topical administration of 0.3 mg 3α-hydroxymasticadienonic acid every 24 h for 15 days increase wound closure by approximately 30%. A similar effect was recorded with 10 mg of an anacardic acid mixture consisting of C15:0, C16:0, C17:0, C19:0, and C19:1 [10]. 

Also, using a chick embryo chorioallantoic membrane model, both compounds exhibited proangiogenic activity [10]. However, the hydroalcoholic extract of the stem bark (ethanol–water 7:3) and masticadienonic acid was inactive in both experimental models. In addition, no bactericidal activity was observed in the main bacteria associated with wounds, such as *P. aeruginosa*, *S. mutans*, *Staphylococcus aureus*, and *Escherichia coli* [10]. 

### 6.6. Hypocholesterolaemic Activity

Subcutaneous administration of hexane extract (100 mg/Kg) diluted in safflower oil (1 mL/Kg) reduced cholesterol levels in Wistar rats by 31%, an effect similar to estrone (15 mg/Kg) used as a positive control [44]. 

## 7. Toxicity

Studies are scarce, but it is generally mentioned that it has low or no acute toxicity. For the aqueous extract of the stem bark, oral administration of 1200 to 2000 mg/kg in BALB/c female mice did not show clinical signs of acute toxicity or lethal effects at 72 h of evaluation [6].

Chloroform–methanol (1:1) extract from the stem bark diluted in tween-80 and administered orally in ICR female mice did not exert toxic effects, registering an LD_50_ > 5000 mg/Kg [68]. In the test with *Artemia salina*, lethality was LC_50_ > 1 mg/Kg, and it did not show mutagenic effects in the Ames test [68].

Oral administration of the hexane extract and anacardic acid mixture in BALB/c female mice (Lorke’s method) resulted in an LD_50_ > 5000 mg/Kg [64]. Moreover, anacardic acid methyl ester was not toxic in the *A. salina* model, and LC_50_ > 1000 ppm [33].

Moreover, 6-nonadecenyl salicylic acid (C19:0) and its methyl ester did not induce chromosomal damage in mouse peripheral blood lymphocytes [45]. Moreover, the anacardic acid mixture at 50 μg/mL did not affect the viability of human peripheral blood mononuclear cells at 24 h [9]. 

## 8. Conservation and Management

*Amphipterygium adstringens* is classified on the IUCN Red List as a *Vulnerable* species [12]. However, a recent study that analyzed current populations placed it within the *risk* category [69].

The stem bark is commercialized both in local and international markets, and it is estimated that more than 57 tons are collected per year in the South-Central region of Mexico, mainly in the State of Morelos [70]. Uncontrolled debarking and excessive trade have put the plant’s survival at risk and have considerably reduced its natural distribution [11]. Similarly, the loss of their habitat due to land-use change, cattle grazing [12,71], climate change [69], and the high percentage of seedless fruits are some of the factors that contribute to the current risk status of the plant [21].

Against this background, some debarking strategies and techniques that can reduce death and increase bark regeneration are suggested [2,11]. However, sustainable use, proper management, and long-term conservation based solely on debarking is complex since bark regeneration depends on many factors, mainly, the sex of the tree, the amount of bark harvested, the depth of the incision, the harvest season, the physiology of the plant, the microclimate, and the concentration of active metabolites [2]. 

It is reported that the stem bark of *A. adstringens* has a high regeneration capacity, although it varies with the sex of the tree and the period of drought and rain [27]. Moreover, the highest regeneration rate occurs in female trees (although their survival is lower than that of males) in the dry season and with a maximum debarking of 50% [2].

It is also recommended not to selectively debark trees with reddish bark because it would affect the demography of the plant [2,27]. However, it should be noted that there are discrepancies between authors on the relationship between bark color and plant sex and the preference of the collectors (Figure 2b,c). In this regard, Solares-Arenas, and Gálvez-Cortes mention that due to the color of the bark, the trees that are being debarked the most and are at risk are male trees [27], while Beltrán-Rodríguez et al. point out that the trees are female [2,71]. However, recent studies show no relationship between the color of the bark and the sex of the tree (personal communication). Therefore, it is essential to continue with ecological and morphological studies of the species to clarify these discrepancies.

Furthermore, there are few studies on chemical composition related to the harvest season and sex of the tree. A preliminary study reported differences in the concentration of the primary terpenes present in stem bark, masticadienonic acid, and 3α-hydroxymasticadienonic acid [72]. A more significant accumulation of terpenes was identified in the stem barks of female plants, with a maximum accumulation of masticadienonic acid in February (0.89% dry weight) and 3α-hydroxymasticadienonic acid in November (0.24% dry weight). Similarly, the presence of terpenes in the exuded latex-resin was identified but not quantified [72]. 

## 9. Patents

A search of patent databases (WIPO, IMPI, 9 May 2022) revealed 15 documents with the terms “cuachalalate”, *Amphipterygium adstringens*, and *Juliana adstringens*.

Regarding its use, patent US-5843421-01/12/1998 describes the preparation of a tonic from the ground stem bark extract of *A. adstringens* (2.5 to 25%) to stimulate hair growth in humans and animals [73]. Document JP-1999302128-02/11/1999 presents the production of a cosmetic with 0.0001 to 20% ethanol extract of stem bark to stop hair loss and counteract dandruff [74]. Additionally, JP-19982453954-14/09/1998 and JP-2000169497-20/06/2000 discloses two triterpenes (6-oxo-3-hydroxyeufa-7,24-dien-26-oic acid and 3,8-dioxo-7-hydroxy-7,9-cyclo-7,8-secoeufa-24-en-26-oico) isolated from the ethanolic extract of the stem bark as inhibitors of testosterone 5-alpha reductase and its use against skin conditions and alopecia [75,76]. Moreover, patent JP-1999209236-03/08/1999 describes a formulation with 0.005 to 20% ethanol extract from leaves, stems, flowers, bark, seeds, and fruits to inhibit testosterone-5-alpha reductase [77].

Document JP-2000198715-18/07/2000 describes the formulation with 0.001 to 20% of an aqueous extract or ethanolic extract of the leaves and stems from inhibiting lipid peroxide formation [78].

Patent MX-350431-05/09/2017 describes the application of organic bark and leaf extracts to prevent and treat skin lesions induced by overexposure to solar radiation [79], while document JP-1999209298-03/08/1999 refers to preparing an ointment, cream, or lotion with 0.1 to 10% ethanol extract from leaves, stems, and flowers to treat dry skin and dermatosis [74]. Patent JP-1999209261-03/08/1999 discloses a formulation with 0.001 to 10% ethanol extract from leaves, stems, and flowers to promote the production of hyaluronic acid and prevent skin aging [76]. Moreover, patent JP-4198260-10/10/2008 describes the preparation of a cosmetic containing *A. adstringens* to whiten the skin and improve its appearance [76].

Patent US-7045156-16/05/2006 explains preparing a solution with *A. adstringens* stem bark and liquid mint to eliminate *H. pylori* [80] and infections caused by stomach parasites, while document MX-2012013361-09/08/2013 describes the use of the hexane extract of the stem bark and anacardic acid mixture as virulence inhibitors in *P. aeruginosa* [81].

Finally, patent US-5869059-09/02/1999 describes a formulation with plants of the genus *Equisetum* and *A. adstringens* to relieve hemorrhoids [82], while document MX-2017004366-09/11/2018 describes a formulation with *A. adstringens* for the treatment of periodontal disease [83], and document MX-MX/a/2008/013281-15/06/2010 discloses a food supplement to reduce fetid odors of secretions and excretions in animals [84].

## 10. Perspectives and Challenges

There are different socio-economic and ecological challenges that must be overcome to protect *A. adstringens* and achieve sustainable management. One of the main problems that stem bark marketing faces is the lack of fair trade because collectors receive low economic remuneration, while intermediaries and industries are the primary beneficiaries [11]. Similarly, there is overexploitation of *A. adstringens*, which, together with other factors, such as the alteration of its habitat by human activity, have enormously reduced the size of its population in only a couple of decades [12]. However, since the tree is distributed in at least four protected areas in Mexico and some conservation efforts have been made, it is urgent to develop strategies for sustainable use, adequate management, and long-term conservation [12]. In this context, it is suggested that commercial plantations could help reduce the pressure on natural populations [24]. However, for more effective use, it is necessary to carry out studies aimed at identifying and quantifying the active compounds in aerial parts (branches) and corroborating their pharmacological efficacy.

Furthermore, a complementary strategy would be to obtain the bioactive compounds identified in *A. adstringens* from other plant species with renewable structures, such as fruits. In this sense, it is known that the primary source of anacardic acids is the cashew nut shell liquid of *Anacardium occidentale* L. [85]. In this regard, it was reported that despite the differences in saturation and chain length of the anacardic acid mixture from *A. occidentale* and *A. adstringens*, the inhibitory activity of quorum sensing was similar [65]. These types of strategies would favor yield by taking advantage of renewable structures and prevent tree death.

Similarly, research has been limited to analyzing the properties of extracts with organic solvents, where terpenes and phenolic compounds have been identified, but none of them have been identified in aqueous extracts (personal communication). Thus, analysis of the chemical composition and action mechanisms of aqueous preparations is a question that needs to be investigated in detail. 

Finally, even though there has been an increase in the number of scientific documents related to the pharmacological properties of *A. adstringens*, to date, preclinical studies remain scarce, and there are no clinical studies. Moreover, although the reports indicate a low or null acute toxicity of the plant, more in-depth studies are necessary to guarantee the safety of the treatments developed. 

## 11. Conclusions

There is evidence of the pharmacological properties of the stem bark and compounds of *A. adstringens*, such as its anticancer, gastroprotective, anti-inflammatory, and antimicrobial capacity. However, to support the claims of healing, anti-virulence, and hypoglycemic properties, it is necessary to expand studies.

Although *A. adstringens* has the potential to cure some of the above diseases, so far, there are no clinical studies to support it.

The oral administration of the aqueous extract is one of the main modes of widespread use, but chemical composition studies are scarce.

The increased demand for the stem bark has put the species’ survival in natural populations at risk. In addition, standardization measures are necessary to guarantee the concentration of active metabolites in the extracts and guarantee their pharmacological efficacy.

Complementary pharmacological studies are needed to determine the routes of administration and those related to the mechanism of action and chronic toxicity.

## Figures and Tables

**Figure 1 plants-11-01766-f001:**
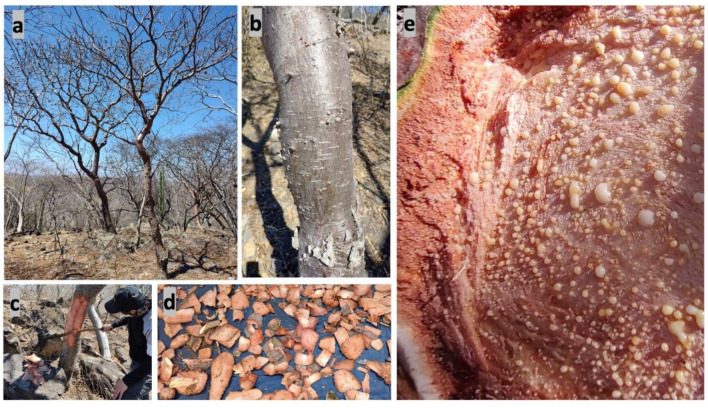
(**a**) *A. adstringens* trees in the dry season; (**b**) stem bark of tree showing smooth outer bark with lenticels; (**c**) traditional debarking; (**d**) traditionally dried stem bark, and (**e**) resin secretion from the stem bark induced by mechanical damage. Pictures taken by J.L. Díaz-Núñez in “Los Sauces” (Tepalcingo, Morelos, Mexico), February 2022.

**Figure 2 plants-11-01766-f002:**
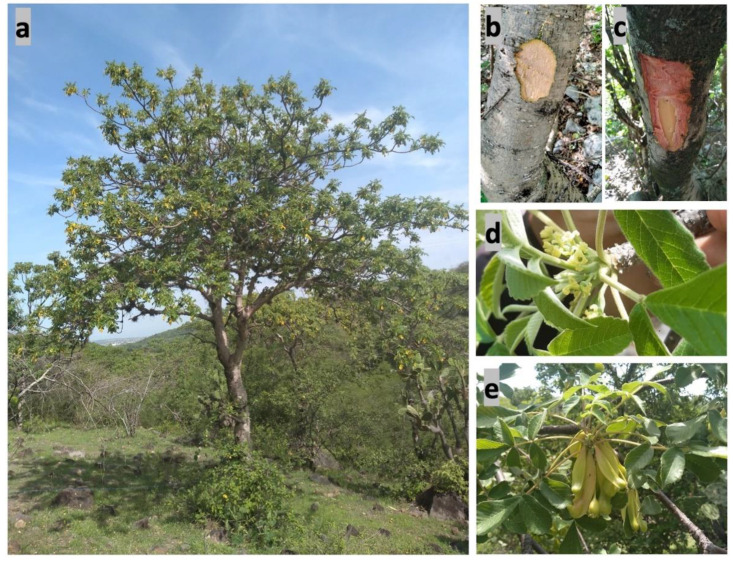
(**a**) *A. adstringens* tree in the rainy season, (**b**) whitish debarked stem commonly associated with female trees, (**c**) reddish debarked stem is associated with male trees, (**d**) female flowers, and (**e**) fruits. Pictures taken by M. Sotelo taken in Cuauchichinola (Mazatepec, Morelos, Mexico), June 2020.

**Figure 3 plants-11-01766-f003:**
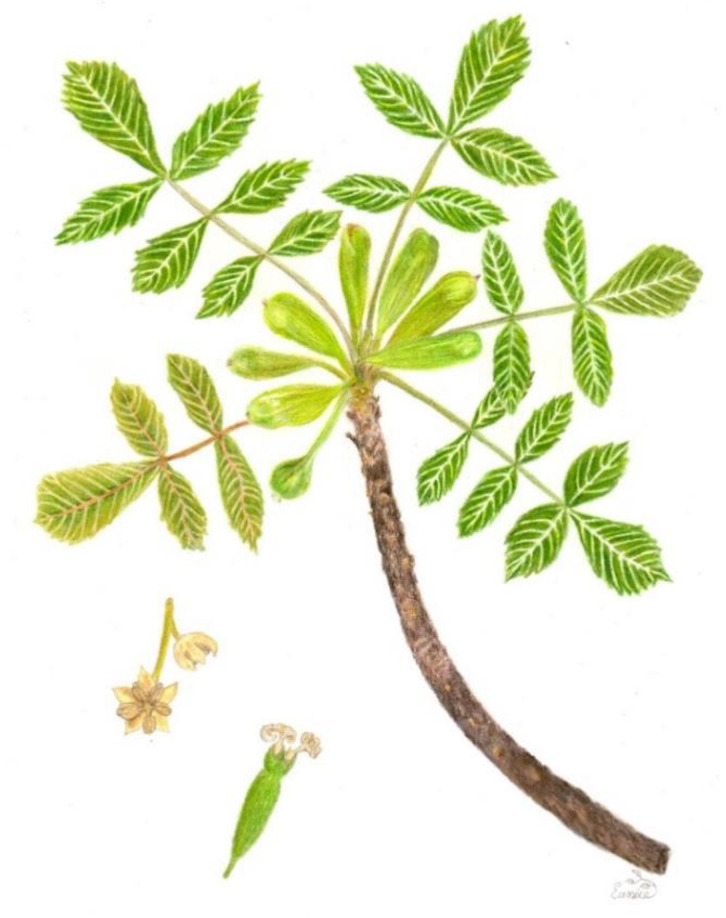
*A. adstringens* sketch showing the structure of the leaves, fruits, and male (left) and female (right) flowers. Color pencil drawing by Eunice Romero, 2022.

**Figure 4 plants-11-01766-f004:**
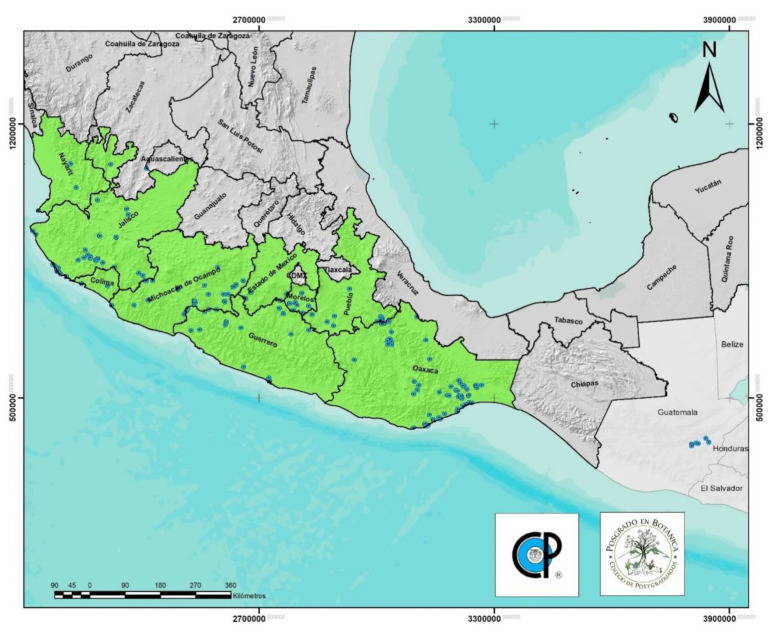
Geographic distribution of *A. adstringens*. The blue dots indicate locations where samples were collected. The World Information Network on Biodiversity (REMIB, CONABIO), W^3^Tropicos^®^-Missouri Botanical Garden Database, National Herbarium of Mexico (MEXU), Herbarium-Hortorio (CHAPA, COLPOS), National School of Biological Sciences (ENCB-IPN), Institute of Ecology, A.C-Xalapa (IE-XAL), Autonomous Metropolitan University (UAMIZ), Luz María Villarreal de Puga Herbarium (IBUG), María Agustina Batalla Herbarium (FCME) and Chapingo Autonomous University Herbarium (CHAP). Qgis version 3.16, OSGeo (Beaverton, USA) software was used. (https://www.qgis.org/es/site/; accessed on 13 June 2022).

**Figure 5 plants-11-01766-f005:**
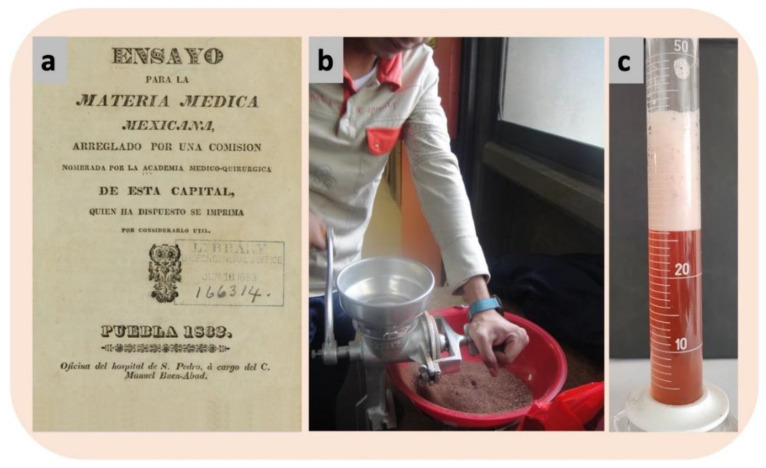
Primary modes of widespread use of the stem bark of *A. adstringens* for the treatment of diseases or injuries. (**a**) *The Essay on Mexican Materia Medica* is one of the first manuscripts where the medicinal use of the plant is reported (https://archive.org/details/61540040R.nlm.nih.gov/page/n3/mode/2up; accessed on 28 June 2022); (**b**) pulverized stem bark and (**c**) boiled aqueous extract of the stem bark with the production of foam (after shaking) that is indicative of the presence of saponins.

**Figure 6 plants-11-01766-f006:**
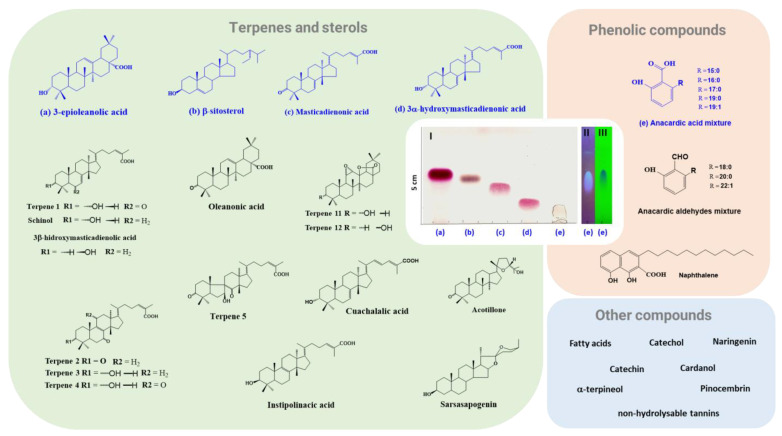
Phytochemicals identified in the stem bark of *A. adstringens*. The most abundant bioactive compounds in the extracts with organic solvents are indicated in blue. (**I**) Thin-layer chromatography (TLC) revealed 10% vanillin in sulfuric acid (hexane/ethyl acetate, 7:3). The TLC plate is stimulated with shortwave (**II**) and longwave (**III**) ultraviolet light, with which the anacardic acid mixture is visualized (hexane/ethyl acetate, 1:1). In the case of the “other compounds” quadrant, these have only been identified by employing HPLC, GC/MS, or qualitative assays. The chromatoplate was made with compounds isolated and characterized by our research group.

**Figure 7 plants-11-01766-f007:**
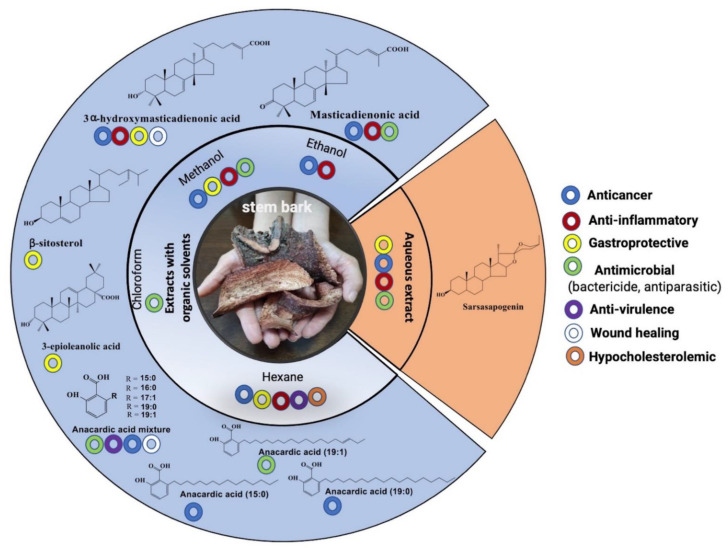
Pharmacological properties of *A. adstringens* stem bark extracts and main bioactive metabolites. Anticancer [5,6,7,42,43,45,49,50,51,52,53,54,55], anti-inflammatory [7,48], gastroprotective [57,58,59,60,61], antimicrobial [5,9,46,47,62,63], anti-virulence [64,65], wound healing [10], and hypocholesterolemic activity [44].

**Table 1 plants-11-01766-t001:** Botanical classification of *A. adstringens*.

Kingdom: Plantae
Sub Kingdom: Tracheobionta
Division: Magnoliophyta
Class: Magnoliopsida
Sub Class: Rosidae
Order: Sapindales
Family: Anacardiaceae
Genus: *Amphipterygium*
Species: *A. adstringens*

## Data Availability

Not applicable.

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
