# Peer review of "Amphipterygium adstringens (Schltdl.) Schiede ex Standl (Anacardiaceae): An Endemic Plant with Relevant Pharmacological Properties"

_plants, 2022, doi:10.3390/plants11131766_

Round 1

Reviewer 1 Report

The paper entitled “Amphipterygium adstringens (Anacardiaceae) is an endemic plant with relevant pharmacological properties” is an interesting, well-planned and structured, consistent review. 

After reading the text, I did not find any weaknesses. Nevertheless, I have a few minor comments and suggestions.

Most of all Figure 6 must be enlarged as it is illegible. In its current form, it can be placed vertically on the entire page or divided into 2 parts or separate figures. Patterns need to be larger and clearer - this is an important part of the work.

Table 1 the species should be: A. adstringens. A. is missing.

I am not sure that figure 3 is necessary.

The title of chapter 3 should be Distribution. Traditional uses and ethnopharmacology is the title of Chapter 4.

Reference [24] on page 5 is missing. I have found it on page 13, line 478 cited the first time.

A. adstringens in Figure 4 and 5 titles should be italics.

References in the brackets should be in numeric order, e.g., Line 200, 332, and others.

Reference 75 is not mentioned in the text at all.

In my opinion, chapter 9 Toxicity should be placed on page 11 between pharmacological properties and conservation and management.

Author Response

Dear reviewer, the authors are grateful for your valuable comments and suggestions for our paper.

Most of all Figure 6 must be enlarged as it is illegible. In its current form, it can be placed vertically on the entire page or divided into 2 parts or separate figures. Patterns need to be larger and clearer - this is an important part of the work.

As indicated by the journal's editorial guidelines, all figures were attached with higher resolution jpg files. The image should not be divided since the chromatoplate relates the two parts, one with the terpenic compounds and the other with the phenolic ones. However, we consider your proposal to place the image vertically on the whole page.

Table 1 the species should be: A. adstringensA. is missing.

The species is only adstringens, "A" refers to the initial of the genus. Therefore, we did not make the suggested change.

I am not sure that figure 3 is necessary.

We consider that figure 3 should be kept since the botanical description, mainly of the male and female flowers, cannot be appreciated in the photographs. Similarly, in this part, advice was requested from an expert in the area (Ph.D. Julien B. Bachelier) and a scientific illustrator for its realization. Also, the editors agree to maintain it for the contribution related to facilitating the interpretation of the morphological characteristics of plants.

The title of chapter 3 should be DistributionTraditional uses and ethnopharmacology is the title of Chapter 4.

The suggested change has been made and is highlighted in yellow in the document.

Reference [24] on page 5 is missing. I have found it on page 13, line 478 cited the first time.

All references were reviewed; in this new version, the reference [24] is on line 126.

adstringensin Figure 4 and 5 titles should be italics.

It was made, and the change is highlighted in yellow.

References in the brackets should be in numeric order, e.g., Line 200, 332, and others.

The suggested change has been made and is highlighted in yellow in the document.

Reference 75 is not mentioned in the text at all.

All references were reviewed; in this new version, the reference [75] is on line 451.

In my opinion, chapter 9 Toxicity should be placed on page 11 between pharmacological properties and conservation and management.

The suggested change has been made and is highlighted in yellow in the document.

Reviewer 2 Report

The article is well structured, argued with adequate bibliography and written in a way that allows easy tracking of the ideas that the authors want to highlight.

However, I identified small mistakes that I marked:

Table 1: Spindales is not correct, should be Sapindales

In Paragraph 3, "This species inhabits warm sub-humid climate.." Inhabits is more appropriate for animals, including humans, that can move from one place to another. You can use "populate" or "grow in", as well as other synonyms.

Page 9: MAP - All abbreviations should be explained. In this case is about mitogen-activated protein kinase (MAPK or MAP)

Paragraph 8, after reference 73:  "While.." As this term implies two subordinate activities or carried out at the same time, I consider that it is not good to be used here. You could use the following: too, as well, besides, in addition, additionally, furthermore, further, moreover, etc. You can also put two sentences in comparative form; in this case use the comma and continue with "while" in lower case. Do not put "while" at the beginning of the sentence if it is not subsequent argued by something opposite. This implies for the entire paragraph.

Author Response

Dear reviewer, the authors are grateful for your valuable comments and suggestions for our paper.

Table 1: Spindales is not correct, should be Sapindales

It was made, and the change is highlighted in yellow.

In Paragraph 3, "This species inhabits warm sub-humid climate.." Inhabits is more appropriate for animals, including humans, that can move from one place to another. You can use "populate" or "grow in", as well as other synonyms.

Suggested changes were made, and the text was revised. All changes are highlighted in yellow.

Page 9: MAP - All abbreviations should be explained. In this case is about mitogen-activated protein kinase (MAPK or MAP)

Suggested changes were made, and the text was revised. All changes are highlighted in yellow.

Paragraph 8, after reference 73:  "While.." As this term implies two subordinate activities or carried out at the same time, I consider that it is not good to be used here. You could use the following: too, as well, besides, in addition, additionally, furthermore, further, moreover, etc. You can also put two sentences in comparative form; in this case use the comma and continue with "while" in lower case. Do not put "while" at the beginning of the sentence if it is not subsequent argued by something opposite. This implies for the entire paragraph.

It was made, and the change is highlighted in yellow.
